# Oxone^®^-Mediated TEMPO-Oxidized Cellulose Nanomaterials form I and form II

**DOI:** 10.3390/molecules25081847

**Published:** 2020-04-17

**Authors:** John P Moore II, Soma Shekar Dachavaram, Shobanbabu Bommagani, Narsimha Reddy Penthala, Priya Venkatraman, E. Johan Foster, Peter A. Crooks, Jamie A. Hestekin

**Affiliations:** 1Ralph E. Martin Department of Chemical Engineering, University of Arkansas, Fayetteville, AR 72701, USA; jpm011@uark.edu; 2Department of Pharmaceutical Sciences College of Pharmacy, University of Arkansas for Medical Sciences, Little Rock, AR 72205, USA; SSDachavaram@uams.edu (S.S.D.); sbommagani@uams.edu (S.B.); NRPenthala@uams.edu (N.R.P.); PACrooks@uams.edu (P.A.C.); 3Material Science and Engineering, Virginia Tech, Blacksburg, VA 24061, USA; priya94@vt.edu (P.V.); johanf@vt.edu (E.J.F.); 4Department of Chemical and Biological Engineering, University of British Columbia, Vancouver, BC V6T1Z3, Canada

**Keywords:** cellulose, nanomaterials, TEMPO, oxone^®^, hydrophilicity, crystallinity

## Abstract

The 2,2,6,6-tetramethylpiperidin-1-oxyl (TEMPO) oxidation of cellulose, when mediated with Oxone^®^ (KHSO_5_), can be performed simply and under mild conditions. Furthermore, the products of the reaction can be isolated into two major components: Oxone^®^-mediated TEMPO-oxidized cellulose nanomaterials Form I and Form II (OTO-CNM Form I and Form II). This study focuses on the characterization of the properties of OTO-CNMs. Nanoparticle-sized cellulose fibers of 5 and 16 nm, respectively, were confirmed through electron microscopy. Infrared spectroscopy showed that the most carboxylation presented in Form II. Conductometric titration showed a two-fold increase in carboxylation from Form I (800 mmol/kg) to Form II (1600 mmol/kg). OTO-CNMs showed cellulose crystallinity in the range of 64–68% and crystallite sizes of 1.4–3.3 nm, as shown through XRD. OTO-CNMs show controlled variability in hydrophilicity with contact angles ranging from 16 to 32°, within or below the 26–47° reported in the literature for TEMPO-oxidized CNMs. Newly discovered OTO-CNM Form II shows enhanced hydrophilic properties as well as unique crystallinity and chemical functionalization in the field of bio-sourced material and nanocomposites.

## 1. Introduction

As one of the most abundant resources on Earth, cellulose has been explored for a myriad of applications. Cellulose is a linear polysaccharide chain with repeated β (1–4) glycosidic linkage of D-glucose monomers [1,2]. While traditionally cellulose can be challenging to work with due to its low solubility in water and most organic solvents, various nanocellulose composites have been found to possess a variety of unique properties (e.g., hydrophilicity, film-forming properties, dispersibility, and mechanical stability). Since their discovery by Herrick and Turbak in 1983, cellulose nanomaterials (CNMs) have found their way into industrial applications, including the production of high-flux membranes, composite coatings, absorbent products, food additives, medical devices, and paper product development [3,4]. Currently, CNM research has grown to an all-time high, with the publisher MDPI for example releasing a special edition dedicated to cellulose nanomaterials last year. This has prompted an increased interest in the development of novel compounds produced from CNMs, as well as innovative approaches to the synthesis and characterization of CNMs [5,6].

In 1995, De Nooy et al. discovered a technique for oxidizing primary and secondary alcohols using the radical 2,2,6,6-tetramethylpiperidin-1-oxyl (TEMPO) oxidation method [7]. Chang and Robyt found the oxidation progress worked on insoluble polysaccharides in 1996, and Isogai and Kato (1998) followed soon after by publishing their results on the oxidation of cellulose using the TEMPO oxidation method and the benefits thereof [8,9]. Recent research has indicated that the surface-limited TEMPO oxidation of cellulose produces nanofibers of cellulose 2–20 nm in width [10,11]. Furthermore, this suspension does not flocculate out of solution due to oxidation, and the resulting charged carboxyl moiety occurs only on the surface of the crystalline particle without effecting crystallinity [10,11,12,13,14,15]. As research in CNMs has developed, standardization for characterization has become well-defined, utilizing parameters such as chemical composition, material properties, and image analysis [6]. These parameters have been used to determine the CNMs produced from various sources of material such as banana and pistachio shells [16,17], and their quality.

Fukuzumi et al. clearly stated that TEMPO oxidation of cellulose facilitated the production of oxidized CNMs while also producing some unwanted by-products, such as aldehydes. Although new approaches for cellulose oxidation and nanocrystal formation have made progress, there is still a need for better methods of CNM production, with reduced reaction times under milder conditions. Oxone^®^ (KHSO_5_) is a non-toxic, highly water-soluble, and cost-effective reagent, which when used with TEMPO forms aldehyde-free TEMPO-oxidized cellulose [1,18]. Recently, reports on the synthesis and chemical characterization of TEMPO-oxidized cellulose and the effect of Oxone^®^ as an intermediary in the oxidation process have shown increased yields with Oxone^®^ under milder reaction conditions to produce two forms of CNMs [1]. While TEMPO, Oxone^®^, and its cellulose constituents have received considerable research attention in the last few years, typically only the major product of the reaction (the insoluble fraction) has been isolated, characterized, and utilized for different material applications [19,20,21].

This study seeks to characterize the properties of Oxone^®^-mediated TEMPO-oxidized cellulose nanomaterials (OTO-CNMs) Form I and Form II. OTO-CNMs were characterized for size and morphology utilizing transmission electron microscopy (TEM). The chemical structure was assessed with Fourier-transform infrared spectroscopy (FTIR), and charge density was evaluated via conductometric titrations. Lastly, material properties were investigated, such as crystallinity and crystallite size with X-ray diffraction (XRD), hydrophilicity using contact angle, and polymer enforcement capabilities using bilateral tensile testing on a polymer doped with OTO-CNM. This report is the first to show the material characterization of the two forms of oxidized cellulose nanomaterial utilizing Oxone^®^ produced under mild reaction conditions and standard heating to produce TEMPO-oxidized cellulose nanomaterial Form I (OTO-CNM Form I) and TEMPO-oxidized cellulose nanomaterial Form II (OTO-CNM Form II) (Figure 1).

## 2. Results & Discussion

### 2.1. TEM Microscopy of Oxone^®^-Mediated TEMPO-Oxidized Cellulose

It was proposed that the two forms of OTO-CNM could be oxidized independently. The reaction conditions afford different levels of carboxylation of the cellulose polymer to produce products with different water solubilities. The increased carboxylation also leads to a further size decrease of the individual fibers. The results of the TEM analysis of images Figure 2a,b show that Form II has a smaller fiber diameter and higher aspect ratio than Form I. The forms have different morphologies. The size of OTO-CNMs Form II may be attributed to its complete oxidation, which also results in the size reduction of the cellulose nanomaterials. As can be seen, OTO-CNM Form II, a more functionalized CNM than OTO-CNM Form I, has a much smaller fiber diameter. Analytical TEM analysis (Figure 2c–e) shows an oxidative cleaving of the original 20-micron-sized microcrystalline cellulosic polymeric material into cellulose nanomaterials.

Typically, mechanical treatment such as homogenization and sonication are necessary as a dispersion step in the formation of cellulose nanomaterials and solutions thereof. Without mechanical treatment, the Oxone^®^-mediated TEMPO oxidation cellulose oxidizing method afforded CNMs exhibiting size parameters of 16 ± 2 nm diameter for Form I down to 5 ± 2 nm for Form II, as illustrated in Figure 2a–c. Furthermore, the length of both OTO-CNM forms was characterized to be between 100 and 200 nm, and the median length of 150 nm was then chosen for the aspect ratio calculations. The aspect ratio (length divided by diameter) was around 9 for Form I and 30 for Form II. Higher aspect ratios for Form II show a greater potential in the reinforcement capabilities when incorporated into composite materials [22]. The important difference regarding the current study, however, is that the fully functionalized Form II product from the TEMPO/Oxone^®^ synthetic procedure requires no mechanical treatment to create smaller nanoparticulates, as illustrated through Appendix A.

### 2.2. FT-IR Spectroscopy of TEMPO-Oxidized Nanocellulose

Figure 1 illustrates the chemical structures of the OTO-CNMs. FT-IR spectroscopic analysis (Figure 3) shows a distinct change in the O–H stretching region around 3300 cm^−1^ between the cellulose starting material and the TEMPO-oxidized cellulose nanomaterial samples. Typically, the 3300 cm^−1^ stretch is attributed to water within the sample. However, to remove the water, the samples were vacuum-dried before FTIR analysis. Furthermore, the O–H stretching region of cellulose associated with the peak at 3300 cm^−1^ is known to be representative of the intermolecular forces exhibited through hydrogen–hydrogen interactions of the O–H groups present on the cellulose [23]. Therefore, the broadening of the O–H stretching frequency may be attributed to the variability introduced by the carboxylate groups and the interactions thereof present in OTO-CNM Form I and more present in Form II. The analysis also shows the presence of a high-intensity peak at 1600 cm^−1^ in the FTIR spectrum of Form II, which is of lower intensity in the spectrum of Form I. A peak at 1600 cm^−1^ demonstrates the presence of the C=O moiety of the carboxylate group in the oxidized cellulose samples. Peak intensity at 1600 cm^−1^ is directly proportional to the carboxyl content in the oxidized cellulose; little absorption can be observed at this wavelength in the FT-IR spectrum of cellulose, while peak intensity increases from Form I to II.

### 2.3. Charge Density Characterization by Conductometric Titration

CNMs produced via an oxidative technique will often contain carboxyl groups bound to them. Utilizing surface titrations, we can quantitatively access the active hydrogen bond sites, i.e., the carboxyl groups present on the surface of the OTO-CNMs. By plotting the conductivity as the dependent variable and the volume of the base as the independent variable, a buffered region can be observed. The number of moles of carboxyl groups can be determined through analysis of the buffered area as shown in Figure 4 and Table 1. Figure 4 shows representative curves obtained from the conductometric titration of microcrystalline cellulose (MCC) (Figure 4a), OTO-CNM Form I (Figure 4b), and OTO-CNM Form II (Figure 4c). OTO-CNM titrations revealed a surface charge of 841 ± 92 and 1620 ± 42 mmols functional group/kg of CNM for Form I and Form II, respectively, as shown in Table 1. The increased surface charge of Form II when compared to Form I can be attributed to the increased carboxylate content of Form II and therefore increased molecular interactions with water. This information would suggest that these anionic charged sodium carboxylate groups in higher densities result in more sufficiently dispersed CNM solutions.

TEMPO-oxidized CNMs are already known to have a relatively high surface charge density that varies depending on the raw materials, with literature values ranging from 200 to 2000 mmol/kg [6,10,24,25,26]. OTO-CNM Form II shows double the carboxylation when compared to Form I, further illustrating the differences in the chemical structures.

### 2.4. XRD Analysis of Oxone^®^-Mediated TEMPO-Oxidized Cellulose

The crystalline structure of CNC produced via synthesis with Oxone^®^ and TEMPO can be better understood through X-ray diffraction or XRD. XRD is a tool used in the characterization of crystallinity and crystallite sizes for cellulose. Utilizing the Rulan–Rietveld approach, we can determine the crystallinity of the CNC (Table 2), which appears to be at the well-defined interval of 2θ of ~14–17°and ~22.7° shown in Figure 5 [6]. These peaks, with the respective planes 101 and 002, are indicative of cellulose crystalline structure as seen in Figure 5, and fall within the reported crystallinity range of 64–91% [27]. Furthermore, crystallite sizes can be investigated using the Williamson and Hall method, and crystalline size can be characterized using the Scherrer equation [28]. Lastly, all raw numbers used for any calculations in Table 1 can be seen in Figure 1.

Analysis of the XRD data revealed an increase in crystallinity for the OTO-CNMs as compared to the control (microcrystalline cellulose, i.e., MCC), as well as crystallite sizes in the range of 1.4–3.3 nm, as shown in Table 2. The OTO-CNMs had a slightly higher crystallinity than the MCC starting material, and Form II showed the highest crystallinity among the samples. Although the percent crystallinity of Form II obtained by subtracting the amorphous peak from the crystalline peak is higher as indicated by XRD, the broad peaks prevent us from concluding that OTO-CNM Form II is more crystalline than Form I. This shows that the oxidation of the cellulose has little effect on the cellulose crystallinity but does affect crystalline heterogeneity. In this XRD analysis, the crystallite size was confirmation that the OTO-CNM Form II produced through Oxone^®^-mediated TEMPO-oxidation formed smaller crystallites of cellulose during the oxidation process than OTO-CNM Form I. XRD raw data used to determine crystallinity and crystallite size is available in Appendix A.

It is important to note that XRD is not well-suited for particle sizes over 20 nm, and the length of the fiber may not have been observed. XRD crystallite calculations were made with the assumption of cubic crystalline structures [29]. While this provides an excellent qualitative assessment for crystallinity, it is not an accurate size measurement. Therefore, other techniques discussed earlier in the article were used as the primary characterization of the cellulose material size and morphology.

### 2.5. Hydrophilicity (Contact Angle) of Oxone^®^-Mediated TEMPO-Oxidized Cellulose

The contact angle data shows contact angles in the range of 16.1° to 58.5° (Figure 6). Typically, materials exhibiting a contact angle below 90° are considered hydrophilic. The material presented exhibits a contact angle of 16–32°, within or below the 26–47° reported in the literature for TEMPO-oxidized CNM (Figure 7) [30,31]. Reported contact angles from TEMPO-oxidized cellulose produced by Benkaddour et al. without the Oxone^®^ intermediary (Figure 7) showed similar hydrophilicity to the OTO-CNM Form I [30]. While Fukuzumi et al. showed the most hydrophilic of the TEMPO-oxidized CNMs reported, neither Baenkaddour et al. nor Fukuzumi et al. showed contact angles as low as those exhibited by OTO-CNM Form II [31]. The TEMPO-oxidized cellulose Form II nanomaterial pushes the limits of hydrophilicity of oxidized cellulose nanomaterials. This increased hydrophilicity allows for the development of more hydrophilic cellulose materials. OTO-CNMs may provide an alternative natural material for the enhancement of hydrophilicity within natural-based polymeric materials in areas such as membrane separation, where improved hydrophilicity can increase membrane performance.

### 2.6. Tensile Testing Characterization

CNMs are known to have robust mechanical properties with tensile moduli, usually ranging from 1 to 10 GPa and going as high as 86 GPa [19]. Furthermore, CNMs are capable of improving the strength of other materials when incorporated as a polymeric nanocomposite [16]. Thereby, tensile tests were performed on cellulose triacetate polymer nanocomposite membranes doped with each of the OTO-CNMs to characterize their mechanical properties. As shown in Table 3, the Form II membrane had the highest values for each of the tested properties, except for tensile strain at maximum load, i.e., extension or elasticity, where no statistical difference was observed between samples. In other words, while the strength of the materials was enhanced, elasticity remained unchanged. Polymeric materials can benefit from the additional crystalline strength and fibrous high-aspect-ratio materials provided as a support structure when incorporated throughout the material matrix [16,17].

## 3. Materials and Methods

### 3.1. General Procedure for the Oxidation of Cellulose Using TEMPO, NaOCl, and Oxone^®^

Cellulose powder, as shown in Figure 7a (1 g, ~20-micron size), was suspended in 0.05 M aqueous NaHCO_3_ solution (50 mL, pH 7.5 to 8.5) in a round-bottomed flask followed by the addition of TEMPO reagent (0.016 g, 0.1 mmol), and Oxone^®^ (0.5 g); then, a 2-M NaOCl solution (0.5 mL) was added to the reaction mixture under a nitrogen gas atmosphere. The temperature of the reaction mixture was raised to 60 °C using an oil bath and stirred at this temperature for 24 h. The reaction mixture was then cooled to ambient temperature, and methanol (4 mL) was added to quench the excess TEMPO reagent. The reaction mass was then filtered through a medium-porosity (10–15 microns) sintered glass funnel, washed with water (100 mL) to remove inorganic materials, and dried under a vacuum to afford partially oxidized cellulose carboxylate (Form I) as an off-white solid (Figure 7b) [1].

To the above filtrate, ethanol (50 mL) was added to afford an immediate white precipitate of TEMPO-cellulose Form II, which was then centrifuged at 10,000 rpm for 30 min. Fully oxidized TEMPO-oxidized cellulose Form II was isolated after centrifugation by separating the supernatant from the pellet. The obtained white solid was washed with aqueous ethanol (ethanol and water 8:2; 3 × 50 mL) to remove inorganic materials and dried under a vacuum to afford OTO-CNM Form II as an off-white solid (Figure 7c) [1].

### 3.2. Solution Preparation of Oxone^®^-Mediated TEMPO-Oxidized Cellulose

Solutions of MCC, OTO-CNM Form I, and OTO-CNM Form II were used as the stock solution. After mixing in a graduated flask, 50 mL aliquots of stock solution were separated. Aliquots were then prepared with and without mechanical treatment. Mechanical treatment involved homogenization in a blender for up to one hour, followed by sonication using probe sonication [32]. A Newtown, CT USA, QSonica, LLC. produced Q120 sonicator was used to create suspensions of the oxidized cellulosic material. These solutions were sonicated for 10 min at 125 watts with 40% efficiency.

### 3.3. Preparation of Oxone^®^-Mediated TEMPO-Oxidized Cellulose Thin Films

The preparation of the OTO-CNM thin films was carried out via two methodologies: the casting-through filtration method, and the wet-casting method (drying) [31,32]. Casting-through filtration was performed by using 50 mL of a 0.1% by volume concentration of OTO-CNMs, and the solution was filtered through a 0.22 micron PVDF hydrophilic filter [19]. The PVDF membrane was dried at 45 °C throughout three days before the OTO-CNM thin film was removed from the surface of the PVDF membrane. The wet casting was carried out by coating aminated glass slides with 0.5 mL of a 0.1% OTO-CNM solution and allowing the slides to dry at room temperature overnight. Wet casting works because electrostatic forces enable the auto-assembly of the polymer thin films as the solvent is removed.

### 3.4. Microscopy

A JEOL JEM-1011, transmission electron microscope Peabody, MA 01960 tunneling electron microscope (TEM) was used to determine the physical properties of the microcrystalline cellulose starting material, and Oxone^®^-mediated TEMPO-oxidized cellulose nanomaterials (OTO-CNMs). The images were analyzed using ImageJ to determine the width of the cellulosic nanomaterials. Microcrystalline cellulose and OTO-CNM Form I and Form II were suspended in distilled water at a concentration of 0.05% by mass. Then, 2.5 microliters of reagent was dropped onto a copper TEM grid. After 24 h of drying, 10 microliters of 1.0% phosphotungstic acid, a negative stain, was used for image clarity of the cellulose fibers [33]. Phosphotungstic acid, when used, can create residual dye spots appearing around the cellulose particles.

### 3.5. Spectroscopy

A PerkinElmer Frontier FTIR-ATR spectrometer from Shelton, CT was used to characterize the chemical differences between the oxidized cellulosic material and traditional microcrystalline cellulose. The FTIR-ATR analysis was done on a single bounce diamond/zinc selenide crystal in the range of 4000–650 cm^−1^ wavenumber with seven repetitions at a resolution of 4 cm^−1^ [6].

### 3.6. Charge Density

To determine the surface charge density of the CNMs, conductometric titrations were carried out. Dispersions of 1 wt% CNMs were corrected to the concentrations of 0.05 M HCl (Sigma Aldrich) and 0.02 M NaCl (Sigma Aldrich). The dispersions were titrated with 0.05 M of NaOH (Sigma Aldrich) at a rate of 0.20 mL/min. Well-distributed dispersions and consistent protonation of the OTO-CNMS in solution were maintained with all samples of OTO-CNM Form I and Form II. Titration conductivity values were determined through plotting against the volume of NaOH added. Since the conductivity is directly proportional to the volume of electrolyte, the measured values of conductivity were corrected using Equation (1) [6].
(1)$$Conductivity_{c}=Conductivity_{m}\left(\frac{V_{i}+V_{0}}{V_{i}}\right)$$

Analyses were performed in triplicate, and equivalence points were obtained from the intersection of the least-squares regression line fit to the data points making up the conductometric titration curve. Blank titrations were used to correct for any free ions.
(2)$$\frac{mmol\;SO^{4-}}{kg\;cellulose}=\frac{C_{NaOH}*V_{NaOH}}{W_{CNC}}$$

Equation (2) was used to calculate the surface charge density of the CNCs [24,25].

### 3.7. X-Ray Diffraction (XRD)

MCC and OTO-CNM Form I and Form II samples were prepared on glass coverslips with the addition of an adhesive enamel. OTO-CNM samples were subjected to X-ray diffraction analysis from a 5–45° angle (2θ) (Figure 5) A Philips Co., Netherlands PW1830 Double System Diffractometer was used to conduct the measurements.

Crystallite size and lattice strain were investigated using the Williamson and Hall method [28]. This method states that the contributions from particle size and strain are independent of each other and have a Cauchy-like profile; the observed line breadth is the sum of the breadth due to crystallite size and lattice strain (Equation (3)) [34].
(3)$$B_{hkl}=B_{Crystalline}+B_{Strain}$$

Crystallite size can be characterized using the Scherrer equation (Equation (4)), where peak width (*B*) is inversely proportional to crystallite size (*L*) [35,36].
(4)$$B_{Crystalline}=\frac{K\lambda }{Lcos\theta }$$
where (*B*) is line broadening at half the maximum intensity (FWHM), (*K*) is a dimensionless shape factor (0.94), (*L*) is the mean size of the ordered crystalline domains, (*λ*) is the wavelength of the incident light, and (*θ*) is the angle of incidence. Lattice strain is a measure of the distribution of lattice constants, such as lattice dislocations arising from crystal imperfections [29].

The crystallinity (*X_c_*) of a CNM sample can be calculated using the total diffracted X-ray intensity assuming the signal comes from both the crystalline and amorphous regions, as shown in Equation (5). This calculation is based on Ruland’s principles and Rietveld analysis, requiring the total crystalline area (*A_c_*) and the total amorphous area (*A_a_*) of the deconvoluted XRD pattern after the subtraction of a background spectrum [6].
(5)$$X_{c}\left(\%\right)=\frac{A_{c}}{A_{c}+A_{a}}\times 100\;\%$$

### 3.8. Contact Angle

For contact angle measurements, the sessile drop technique was used. Here, 0.2 mL of water was suspended from the tip of a needle [37]. Then a platform was raised until the drop was released from the needle and attached to the surface of the material being tested. The entire procedure was videographed using a high-speed camera mounted on the instrument. Contact angle measurements were then calculated using the video footage post-experiment. Hydrophobic or hydrophilic properties were measured using the contact angle [37].

### 3.9. Preparation of Polymeric Nanocomposite Film

To increase the nanodispersion of OTO-CNM Form I and Form II for the production of polymeric nanocomposite films, sonication was used [32]. To prepare the film solution for sonication, a mixture of the OTO-CNMs (1 wt%) and N-methyl pyrrolidone solvent (90 wt%) was mixed thoroughly. The solution was loaded into the sonication apparatus and placed in an ice bath with the sonicator probe inserted into the center of the solution. The loaded apparatus was properly contained, and the QSonica Q120 sonicator run settings were entered (500 watts with 40% η @ 20 kHz, 20 s on/10 s off for a run time of 15 min). Post sonication, the solution was stored at room temperature.

CTA, from Millipore Sigma (also known as Sigma-Aldrich) was used. Cellulose polymer (10 wt%) was added into NMP to complete a 100 g membrane casting solution [38]. Reagent bottles contained the three different solutions from the sonication protocol (1 wt% OTO-CNMs, 9 wt% CTA, and 90 wt% NMP) for film casting; further characterization can be identified in Table 4 below.

Once the final combination of film materials was completed, the mixture was sealed in the reagent bottle to avoid the evaporation of the solvent. The bottle was continuously mixed in a bottle roller over 72–120 h at room temperature. All films were generated by non-solvent induced phase separation [39]. Sample nanocomposite films and the control CTA films were cast from the same batches of each respective solution composition shown in Table 1. The film solution was evenly poured onto the glass plate and spread out using a calibrated Gardco Microm II film application casting knife. The glass plate was quickly placed into the deionized water bath kept between 49 and 51 °C for 30 min for phase inversion to occur, and consequently, annealing took place in a separate DI bath at 87–89 °C for 5 min. All the film properties presented below were carried out in triplicate on film samples extracted from an original sheet from which the test cell samples were also cut. Individual film samples were not used in any other characterization test.

## 4. Conclusions

In this study, we investigated the material properties of the two unique forms of CNMs produced via the Oxone^®^-mediated TEMPO-oxidation process (OTO-CNM Form I and OTO-CNM Form II). Cellulose structures that fall within the scope of CNMs were identified and were shown to develop using the Oxone^®^-mediated TEMPO-oxidation process with and without mechanical treatment. The facile synthesis of CNMs in mild conditions without additional treatment allows for advances in the development of industrially safe production of CNM. Unique chemical structures of the two fractions of OTO-CNM qualitatively identified through FTIR showed the carboxylation of the OTO-CNMs, as illustrated in Figure 1. Quantitatively, OTO-CNM Form II showed twice the amount of surface charge present on the surface than OTO-CNM Form I. Carboxylate functional groups are easily converted to any number of moieties. The increased quantity of carboxyl functional groups on the OTO-CNMs allowed for tunability in the downstream processing and derivatization of CNMs. Nanoscopic crystallite size and crystalline structural properties indicative of native cellulose crystalline structures before and after oxidation were characterized by XRD. XRD showed that the oxidation of the cellulose had little effect on the cellulose crystallinity but did affect crystalline heterogeneity. The contact angle showed that the films created with Form II provided incredibly hydrophilic surfaces. Hydrophilicity in materials has been shown to enhance the longevity of polymeric materials in separation by lowering the fouling. OTO-CNMs may provide a natural polymeric route for the increased hydrophilicity of polymer compounds. Tensile testing showed the reinforcement capabilities of the OTO-CNMs in polymeric materials. OTO-CNMs showed strong mechanical properties without a change in elasticity. Increased strength in natural polymers is a desirable feature in textile and paper manufacturing, and OTO-CNMs can provide that additional strength. Whether used for biological applications or food packaging, the increased chemical tunability of OTO-CNMs in the material design shows continued progress towards effectiveness and usability.

## 5. Patents

A US patent has been submitted with a potential royalty stream to the inventors. International Publication Number WO 2019/023702 A1. International Publication Date: 31 January 2019.

## Figures and Tables

**Figure 1 molecules-25-01847-f001:**
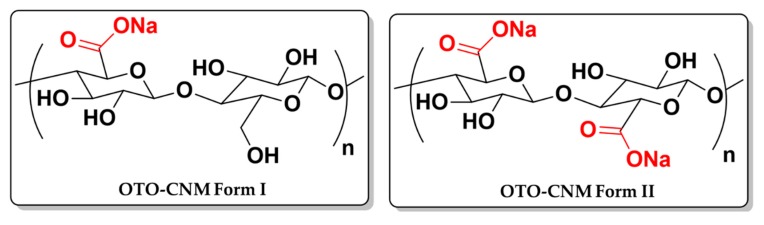
Chemical structures of the two forms of cellulose produced using 2,2,6,6-tetramethylpiperidin-1-oxyl (TEMPO) oxidized in the presence of Oxone^®^ (KHSO_5_): Oxone^®^-mediated TEMPO-oxidized cellulose nanomaterial Form I (OTO-CNM Form I), and Oxone^®^-mediated TEMPO-oxidized cellulose (OTO-CNM Form II) [1].

**Figure 2 molecules-25-01847-f002:**
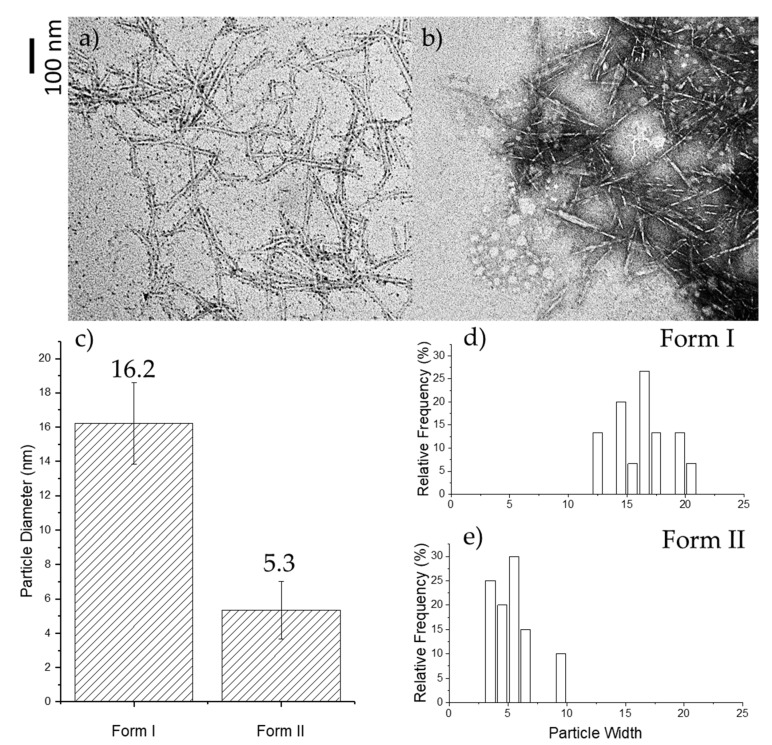
TEM imaging is illustrated in the top left panel for Oxone^®^-mediated TEMPO-oxidized cellulose nanomaterial Form I, OTO-CNM Form I (**a**), and in the top right panel for Oxone^®^-mediated TEMPO-oxidized cellulose nanomaterial Form II, OTO-CNM Form II (**b**) at 100,000 times magnification at an HV (High Voltage) of 100 kV. Below in panel (**c**) the averages of the histograms are graphed, as are histograms of the particle count relative frequency for Form I (**d**) and Form II (**e**).

**Figure 3 molecules-25-01847-f003:**
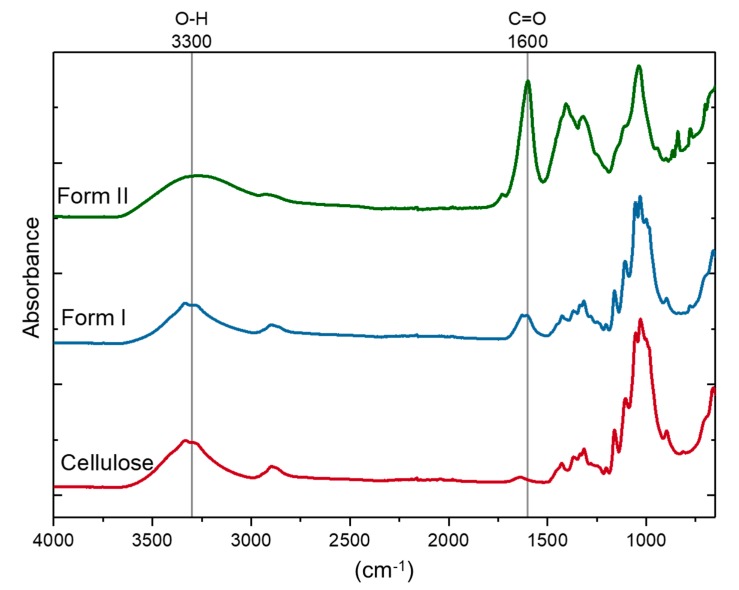
Comparative FTIR spectroscopic analysis of the cellulose starting material and Oxone^®^-mediated TEMPO-oxidized cellulose nanomaterials Form I and Form II.

**Figure 4 molecules-25-01847-f004:**
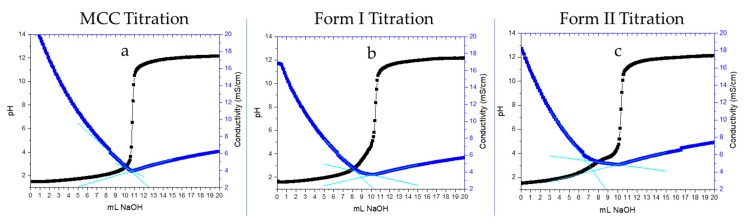
Representative conductometric titration curves plotting corrected conductivity vs. volume of titrant for (**a**) HCL blank, (**b**) Oxone^®^-mediated TEMPO-oxidized cellulose nanomaterial Form I, and (**c**) Oxone^®^-mediated TEMPO-oxidized cellulose nanomaterial Form II.

**Figure 5 molecules-25-01847-f005:**
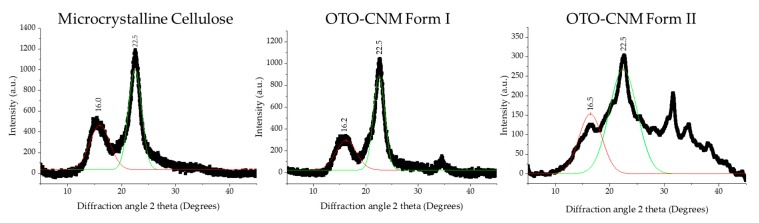
X-ray diffraction (XRD) measurements of OTO-CNMs Form I, Form II, and microcrystalline cellulose (MCC). MCC: microcrystalline cellulose. OTO-CNM Form I: Oxone^®^-mediated TEMPO-oxidized cellulose nanomaterial Form I. OTO-CNM Form II: Oxone^®^-mediated TEMPO-oxidized cellulose nanomaterial Form II.

**Figure 6 molecules-25-01847-f006:**
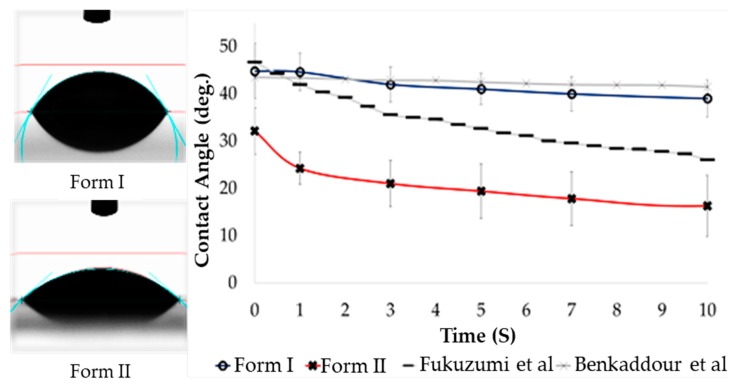
Representative contact angle bubble images on the left, and relative hydrophilicity of OTO-CNM Form I and OTO-CNM Form II as measured through contact angle on the right, sample size 7. MCC: microcrystalline cellulose. OTO-CNM Form I: Oxone^®^-mediated TEMPO-oxidized cellulose nanomaterial Form I. OTO-CNM Form II: Oxone^®^-mediated TEMPO-oxidized cellulose nanomaterial Form II.

**Figure 7 molecules-25-01847-f007:**
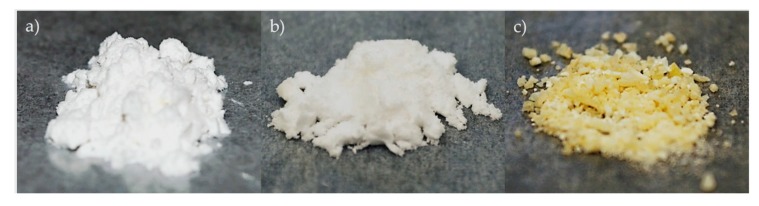
MCC starting material images on the left (**a**), OTO-CNM Form I in the middle (**b**), and OTO-CNM Form II on the right (**c**). MCC: microcrystalline cellulose. OTO-CNM Form I: Oxone^®^-mediated TEMPO-oxidized cellulose nanomaterial Form I. OTO-CNM Form II: Oxone^®^-mediated TEMPO-oxidized cellulose nanomaterial Form II.

**Table 1 molecules-25-01847-t001:** The surface charge density of the cellulose nanomaterials calculated through conductometric titration. MCC: microcrystalline cellulose. OTO-CNM Form I: Oxone^®^-mediated TEMPO-oxidized cellulose nanomaterial Form I. OTO-CNM Form II: Oxone^®^-mediated TEMPO-oxidized cellulose nanomaterial Form II.

Sample	Mmols Functional Group/kg
MCC OTO-CNM Form I	31 ± 10 841 ± 92
OTO-CNM Form II	1620 ± 42

**Table 2 molecules-25-01847-t002:** Crystallinity and crystallite size of MCC, OTO-CNM Form I, and Form II. MCC: microcrystalline cellulose. OTO-CNM Form I: Oxone^®^-mediated TEMPO-oxidized cellulose nanomaterial Form I. OTO-CNM Form II: Oxone^®^-mediated TEMPO-oxidized cellulose nanomaterial Form II.

Sample	Crystallite Size (nm)	Crystallinity (%)
MCC	2.95	59.4
Form I	3.29	63.8
Form II	1.41	67.7

**Table 3 molecules-25-01847-t003:** Average tensile measurements of membrane samples. Tensile measurements were taken using the lateral displacement technique under ASTM D638 standards. By ANOVA statistical analysis, all means are not equal, i.e., F > F critical and *p*-value < 0.05. CTA: Cellulose Triacetate. OTO-CNM Form I: Oxone^®^-mediated TEMPO-oxidized cellulose nanomaterial Form I. OTO-CNM Form II: Oxone^®^-mediated TEMPO-oxidized cellulose nanomaterial Form II.

Sample	Tensile Strain (%)	Tensile Stress (GPa)	Tensile Modulus (GPa)	Maximum Load (kN)
CTA	5.89 ± 2.12	0.44 ± 1.14	0.77 ± 0.43	2.64 ± 0.19
Form I	4.28 ± 1.05	0.84 ± 2.00	2.39 ± 1.15	5.07 ± 0.33
Form II	4.78 ± 1.64	1.21 ± 1.21	2.60 ± 0.58	6.75 ± 0.22

**Table 4 molecules-25-01847-t004:** The compositions of CTA/OTO-CNM membranes based on wt% for Form I and Form II. CTA: Cellulose Triacetate. NMP. N-methyl pyrrolidone. OTO-CNM Form I: Oxone^®^-mediated TEMPO-oxidized cellulose nanomaterial Form I. OTO-CNM Form II: Oxone^®^-mediated TEMPO-oxidized cellulose nanomaterial Form II.

Membrane	Form I (wt%)	Form II (wt%)	CTA (wt%)	NMP (wt%)
Control	0.0	0.0	10.0	90.0
Form I	1.0	0.0	9.0	90.0
Form II	0.0	1.0	9.0	90.0

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
