# Peer review of "Oxone®-Mediated TEMPO-Oxidized Cellulose Nanomaterials form I and form II"

_molecules, 2020, doi:10.3390/molecules25081847_

Round 1

Reviewer 1 Report

The paper describes the characterization of two forms of nanomaterials obtained by the oxidation of cellulose with a mixture of TEMPO and oxone.

This is a very well-written paper with a clear structure and clear objective. Previous works are well described in the introduction and the reader can easily identify the new results reported in this paper (mainly related to the more carboxylated form of cellulose).  The characterization is very clear. This paper is interesting in the field of bio-sourced material and nanocomposites. I recommend publishing this paper.

I have just one very minor remark. Indeed, the same reference was cited twice (references 18 and 19)

Author Response

In response to your comment,

"I have just one very minor remark. Indeed, the same reference was cited twice (references 18 and 19)"

In line 412-413 the duplicate reference was removed and replaced. Correspondingly, references in line 59 and 65 were altered.

Reviewer 2 Report

General comments:

The properties of nanocellulose such as mechanical properties, film-forming properties, and viscosity would make it a feasible material for many applications. TEMPO-oxidized cellulose nanofibril is one of the major nanocellulose, and it is recently reported the effect of Oxone on increase yields under milder conditions. The authors described that the fundamental characteristics of the two-types of the Oxone® mediated TEMPO-oxidized cellulose nanomaterials. Many researchers would interest in this study. In the present state, however, this manuscript is inappropriate for publication.

Individual comments:

(pg.2, ln.74-75) The chemical structure….nuclear magnetic resonance (NMR)…

>There is no description of NMR in this manuscript.

(Figure 2)

>The caption does not match the layout of the figure.

>The contrast of TEM image (Fig. 2a) is quite low and the fiber morphology is very difficult to understand. Further, there are a lot of black dots. What are those?

>How about fiber length? The fiber length is an important factor on physical properties of composites.

>Please confirm HV value.

(Degree of polymerization (DP))

>The reduction of DP is often a problem in the TEMPO oxidation of cellulose, but what about this reaction using Oxone?

(Figure 3)

> Why are bands below 1500 cm-1 in FTIR spectrum of Form II significantly different from spectra of MCC and Form I?

>The OH stretching absorption bands around 3300 cm-1 are derived from hydroxyl groups of cellulose, which is sensitive to the state of hydrogen bondings. I think that this point should be discussed. For example, see the following paper: 10.1023/A:1018448109214.

(pg.6, ln.142: surface charge density)

Surface charge density is depending on raw materials for TOCNs. It is already reported that surface charge density of TOCNs is higher than 1500 mol/kg. Please check the reports from Isogai and Saito (ex. Ref. 10).

(Figure 5: XRD)

> I think that the assignment of diffraction peaks and peak fitting should be considered again. Generally, there are two peaks 110 and 1-10 around 16 deg in XRD of natural crystalline cellulose (cellulose I structure). And the peak at 22.5 deg should attribute to 200 plane. In addition, the peak fitting for the XRD of FormII is less accurate. There are a lot of peaks other than cellulose in FormII. What are those peaks from?

(Table 3)

>I recommend to show s-s curves also. Tensile stress (Tensile strength?) of Form II listed in Table 3 seems to be so high. How high is it compared to other literature?

(Materials and methods)

(pg.9, ln.215)

>The information of pore size of glass funnel used should be added.

(pg.9,ln.231)

>I could not understand “emulsification” procedure (ex. oil used and oil phase content).

(Table 4)

>I could not understand of difference between Control and FormI.

Author Response

Individual comments:

(pg.2, ln.74-75) The chemical structure….nuclear magnetic resonance (NMR)…

>There is no description of NMR in this manuscript.

This is correct, this is a typo and mention of NMR in pg.2, ln.73-75 was removed accordingly. NMR was removed from this manuscript and placed in a sister manuscript titled, “A Facile Microwave Assisted TEMPO/NaOCl/Oxone (KHSO5) Mediated Micron Cellulose Oxidation Procedure: Preparation of Two Nano TEMPO‐Cellulose Forms”. (Reference 1)

(Figure 2)

>The caption does not match the layout of the figure.

The Figure caption in pg. 4 lines 104-106 was adjusted.

>The contrast of TEM image (Fig. 2a) is quite low and the fiber morphology is very difficult to understand. Further, there are a lot of black dots. What are those?

The original images were revisited. Higher fidelity images were produced, and lower magnification (100000x instead of 200000x) were chosen as better representations of the material characteristics. (pg. 4 line 103)

Phosphotungstic acid, when used, can create residual dye spots appearing around the cellulose particles. This comment has been added to the microscopy section 3.4 page 10 line 257

>How about fiber length? The fiber length is an important factor on physical properties of composites.

The length of both OTO-CNM forms was characterized to be between 100-200 nm (pg. 3 lines 97-98)

 No statistical difference was observed between the two sample lengths of the different forms; therefore the median value of 150nm was selected for length-based calculations.

>Please confirm HV value

HV is now confirmed in caption pg. 4 line 105

(Degree of polymerization (DP))

>The reduction of DP is often a problem in the TEMPO oxidation of cellulose, but what about this reaction using Oxone?

 DP, as well as Molecular weight (MW), can be measured in our labs, but right now, we have had trouble accessing the equipment and resources needed to run the experiments promptly.

DP & MW data will not be able to be included in this publication due to laboratory shutdowns.

(Figure 3)

> Why are bands below 1500 cm-1 in FTIR spectrum of Form II significantly different from spectra of MCC and Form I?

The peaks below 1500 cm-1 in MCC, Form I and Form II are mainly due to the C-O stretching, C-H stretching and C-H bending vibrations. Furthermore, in the more oxidized Form II, the percentage of carboxylic content from the additional repeated glucuronic acid unit in the polymeric chain is higher, naturally, Form II may have different from spectra of  MCC and Form I.

>The OH stretching absorption bands around 3300 cm-1 are derived from hydroxyl groups of cellulose, which is sensitive to the state of hydrogen bonding. I think that this point should be discussed. For example, see the following paper: 10.1023/A:1018448109214.

Utilizing your reference the following statement was discussed, “Furthermore, the O-H stretching region of cellulose associated with the peak at 3300 cm-1 is known to be representative of the intermolecular forces exhibited through hydrogen-hydrogen interactions of the O-H groups present on the cellulose [26]. Therefore, the broadening of the O-H stretching frequency may be attributed to the variability introduced by the carboxylate groups and the interactions thereof present in OTO-CNM Form I and more present in Form II.” Pg. 4 line 112-116

(pg.6, ln.142: surface charge density)

Surface charge density is depending on raw materials for TOCNs. It is already reported that surface charge density of TOCNs is higher than 1500 mol/kg. Please check the reports from Isogai and Saito (ex. Ref. 10).

The reviewer’s statement is correct, and the manuscript has been updated in lines 146-147 to address the above statement. i.e.

TEMPO-oxidized CNCs are already known to have a relatively high surface charge density that vary depending on the raw materials, with literature values ranging from 200-2000 mmol/kg [6,10,23-25].

(Figure 5: XRD)

> I think that the assignment of diffraction peaks and peak fitting should be considered again. Generally, there are two peaks 110 and 1-10 around 16 deg in XRD of natural crystalline cellulose (cellulose I structure). And the peak at 22.5 deg should attribute to 200 plane. In addition, the peak fitting for the XRD of Form II is less accurate. There are a lot of peaks other than cellulose in Form II. What are those peaks from?

We appreciate the reviewer's comments in this area and based on literature review Diffraction patterns of CMNs of well-defined peaks assigned to the (002) plane at 2θ appear at approximately 22.7 degrees and the (101) and (101 prime) planes at 2θ  theta appear at approximately 14-17 degrees.  (Reference 6. Foster, E.J.; Moon, R.J.; Agarwal, U.P.; Bortner, M.J.; Bras, J.; Camarero-Espinosa, S.; Chan, K.J.; Clift, M.J.; Cranston, E.D.; Eichhorn, S.J. Current Characterization Methods for Cellulose Nanomaterials. Chem. Soc. Rev. 2018, 47, 2609-2679.)

CNC diffraction patterns for hydrolysis and oxidations at different temperatures vary in crystalline peak intensity and width, leading to different crystallinity and crystallite size, but are otherwise similar in the overall response. Furthermore,  contributions for the (040) plane at 2θ appears approximately at 34.3 degrees.  Some of the additional peaks we don’t exactly know what to attribute them to, right now some of the peaks additional peaks we haven’t been able to attribute to specific structural content based on our analysis or literature references. This will be the topic of future scientific studies.               

(Table 3)

>I recommend showing s-s curves also. Tensile stress (Tensile strength?) of Form II listed in Table 3 seems to be so high. How high is it compared to other literature?

We cannot show s-s curves as they were not saved at the time of analysis. 5-10 samples were run per analysis. Current reports show ranges from 1-10 GPa for films produced by nanocellulose but with some stiffer starting material films as high as 86 GPa have been reported. This has now been addressed in the manuscript in Pg. 8 lines 199-201

(Materials and methods)

(pg.9, ln.215)

>The information of pore size of glass funnel used should be added.

 Addressed pg. 9 line 220

The reaction mass was then filtered through a medium porosity (10-15 microns) sintered glass funnel…

(pg.9,ln.231)

>I could not understand “emulsification” procedure (ex. oil used and oil phase content).

 I should use the correct term “sonication and suspension” to describe the solid phased particles dispersed in the liquid phase water content.  Fixed in Pg. 9 ln 236-237

“Mechanical treatment involved homogenization in a blender for up to one hour, followed by sonication using probe sonication.  A QSonica Q120 sonicator was used to create suspensions of the oxidized cellulosic material.”

(Table 4)

>I could not understand of difference between Control and Form I.

Table 4 had a typo showing no difference between the control and Form I. The correct polymer composition was made clear in pg. 11, line 318, and 322 table 4.

Reviewer 3 Report

Comments for the Authors

In the article, Oxone® mediated TEMPO-oxidized cellulose 3 nanomaterials Form I & Form II, John P Moore focuses on the characterization of the properties of OTO-CNMs. Nanoparticle sized cellulose fibers were confirmed through electron microscopy, and infrared spectroscopy technique was used to determine the chemical structure. Material properties of the OTO-CNMs show cellulose crystalline structural morphologies, as shown through XRD.

The study is interesting however; some issues need to be clarified before this can be considered for publication. The quality of this manuscript may be improved after a minor revision.

The specific comments from this reviewer are given below.

  1. The abstract is descriptive and qualitative. Normally an abstract should state briefly the purpose of the study undertaken and meaningful conclusions based on the obtained results. Hence, this needs rewriting. I would expect brief, yet concise, the quantitative data description of the results in the abstract.
  2. The novelty of the study should be clearly highlighted in the manuscript at the end of the introduction section, as there are some existing literature reports.
  3. Both introduction and discussions should be improved with novelty. Try to include very recent references (2019 to 2020).
  4. The methodological details are not complete and missing protocol description – low reproducibility. Each protocol should be supported with an appropriate reference.
  5. Referencing is not right. Literature needs to be updated with care. At least 20% references should be from recent years 2017-2019.
  6. Editorial issues: The Latin names and Greek letters should be presented in italic in the whole manuscript, the unit presentation should be unified in the whole manuscript, abbreviations presentation should be unified.

Author Response

The specific comments from this reviewer are given below in order asked along with our response.

1. The abstract is descriptive and qualitative. Normally an abstract should state briefly the purpose of the study undertaken and meaningful conclusions based on the obtained results. Hence, this needs rewriting. I would expect brief, yet concise, the quantitative data description of the results in the abstract.

The abstract was almost completely rewritten (pg. 1 lines 14-26)

Abstract: The TEMPO oxidation of cellulose, when mediated with Oxone® (KHSO5), can be performed simply and under mild conditions. Furthermore, the products of the reaction can be isolated into two major components: Oxone® mediated TEMPO-oxidized cellulose nanomaterials Form I and Form II (OTO-CNM Form I & Form II) [1]. This study focuses on the characterization of the properties of OTO-CNMs. 5 and 16 nm Nanoparticle sized cellulose fibers were confirmed through electron microscopy for Form II and Form I respectably. Infrared spectroscopy showed the most carboxylation present in Form II. Conductometric titration showed a two-fold increase in carboxylation from Form I (800 mmol/kg) to Form II (1600 mmol/ kg). OTO-CNMs showed cellulose crystallinity in the range of 64-68 % and crystallite sizes of 1.4-3.3 nm, as shown through XRD. OTO-CNMs show controlled variability in hydrophilicity with contact angles ranging from 16-32 degrees, which is within or below the 26-47 degrees reported in the literature for TEMPO-oxidized CNMs.  Newly discovered OTO-CNM Form II shows enhanced hydrophilic properties as well as unique crystallinity and chemical functionalization in the field of bio-sourced material and nanocomposites. 

2. The novelty of the study should be clearly highlighted in the manuscript at the end of the introduction section, as there are some existing literature reports.

The novelty is now highlighted at the end of the introduction section, line 76-80.

3. Both introduction and discussions should be improved with novelty. Try to include very recent references (2019 to 2020).

Reference 19 added (pg. 15 line 419-420) 2020

Reference 33 added (pg. 15 line 453-454) 2020

Reference 39 added (pg. 15 line 467-469) 2019

4. The methodological details are not complete and missing protocol description – low reproducibility. Each protocol should be supported with an appropriate reference.

Appropriate references were added in lines, Modifications in methods include: 223 & 229 for general oxidation procedure, 237 for solution preparation, 244 for thin film preparation, 257 for TEM microscopy using phosphotungstic acid,  263 for FTIR, REF 32, 38, & 39 line 311, 318, & 326 respectably

5. Referencing is not right. Literature needs to be updated with care. At least 20% references should be from recent years 2017-2019.

The references have been updated, and now 12 of the 39 references are from 2017-2020 (31%)

6. Editorial issues: The Latin names and Greek letters should be presented in italic in the whole manuscript, the unit presentation should be unified in the whole manuscript, abbreviations presentation should be unified.

Abbreviations and corrections were made to lines 309-311, 316, 319, 322, 327,331-333 for continuity

Round 2

Reviewer 2 Report

I have just one minor point to check.

>>Please confirm HV value

>HV is now confirmed in caption pg. 4 line 105

Isn't HV of 100000 kV correctly 100 kV?